# Improvement of Thermal Cycling Resistance of Al_x_Si_1−x_N Coatings on Cu Substrates by Optimizing Al/Si Ratio

**DOI:** 10.3390/ma12142249

**Published:** 2019-07-12

**Authors:** Alexey Panin, Artur Shugurov, Marina Kazachenok, Victor Sergeev

**Affiliations:** 1Institute of Strength Physics and Materials Science of Siberian Branch of Russian Academy of Sciences, 634055 Tomsk, Russia; 2School of Nuclear Science & Engineering, National Research Tomsk Polytechnic University, 634050 Tomsk, Russia

**Keywords:** Al_x_Si_1−x_N coatings, copper, magnetron sputtering, thermal cycling performance, scanning electron microscopy, cracking

## Abstract

The effect of the elemental composition of Al_x_Si_1−x_N coatings deposited on Cu substrates by magnetron sputtering on their structure, mechanical properties and thermal cycling performance is studied. The coatings with Al-Si-N solid solution, two-phase (Al_x_Si_1−x_N nanocrystallites embedded in the Si_x_N_y_ tissue phase) and amorphous structure were obtained by varying Al/Si ratio. It is shown that polycrystalline coatings with a low Si content (Al_0.88_Si_0.12_N) are characterized by the highest thermal cycling resistance. While the coatings with a high and intermediate Si content (Al_0.11_Si_0.89_N and Al_0.74_Si_0.26_N) were subjected to cracking and spallation after the first cycle of annealing at a temperature of 1000 °C, delamination of the Al_0.88_Si_0.12_N coating was observed after 25 annealing cycles. The Al_0.88_Si_0.12_N coating also exhibited the best barrier performance against copper diffusion from the substrate. The effect of thermal stresses on the diffusion barrier performance of the coatings against copper diffusion is discussed.

## 1. Introduction

Due to their high thermal conductivity and excellent corrosion resistance, Cu and its alloys are extensively used in many heat exchange applications. In particular, copper is commonly utilized as a substrate for solar selective absorbers in solar thermal collectors [1,2], while Cu alloys are widely used as combustion chamber liner materials in rocket engines [3]. During operation in these applications, copper components suffer from extremely high thermal loads. For example, in concentrating solar power systems with gas-phase central heat receivers operating temperature can reach 1000 °C [4]. Similar operating temperatures are required in the combustion chamber of rocket engines to enhance their performance [5].

High temperatures induce rapid degradation of copper components for two main reasons. First, copper suffers from easy oxidation with the formation of thick brittle oxide layers [6,7]. Cyclic thermomechanical loads typical for heat exchange applications result in cracking and spallation of the oxide scale that causes the failure of the copper components [5,6,8]. Second, at elevated temperatures copper atoms become mobile, which induces their upward diffusion when Cu-based materials are used as substrates in multilayer structures [9]. As a result, Cu atoms penetrate through above layers and react with oxygen to form copper oxide hillocks on the surface [1,2]. This mechanism leads to material depletion and the formation of Kirkendall voids in the copper substrate causing porosity and loss of strength [1]. Therefore, protective barrier coatings with high thermal stability and oxidation resistance are needed to prevent rapid high-temperature degradation of the Cu components [1,5,10]. These coatings should also have a dense structure to hinder outdiffusion and oxidation of copper substrate material as well as sufficiently high thermal conductivity to provide efficient heat transfer that is necessary for heat exchange applications.

In the last two decades, nanocomposite coatings based on nitrides of transition metals (Ti, V, Cr, Zr, Nb, etc.) have been extensively used due to their high thermal and chemical stabilities as well as increased mechanical characteristics, which are of particular importance in the case of thermomechanical loads [11]. Ternary, quaternary and multielement (high entropy alloy) nitride coatings are characterized by even more improved properties [12,13,14,15,16,17]. In particular, the Al-containing nitride coatings possess increased oxidation resistance at high temperatures [18,19]. Introduction of Si in these coatings gives rise to the formation of the amorphous Si_3_N_4_ grain boundary phase that impedes the growth of metal nitride grains. As a result, the coatings contain metal nitride nanocrystallites embedded in an amorphous silicon nitride matrix [20,21,22]. Such coatings keep their high thermal stability and oxidation resistance at temperatures above 1000 °C [23,24,25,26] and their hardness can exceed 60 GPa [27]. In view of considerable improvement of thermomechanical properties of transition metal nitride coatings by doping with Al or Si, coatings based on the Al-Si-N system have been proposed [28,29,30,31,32,33,34,35]. Thermal stability and high oxidation resistance of Al_x_Si_1−x_N coatings hold at temperatures as high as 1200 °C [31], and the hardness reaches 35 Gpa [35]. Moreover, thermal conductivity of AlN can reach 80% of that of Cu [36], which makes AlN-based coatings promising to protect copper components used for heat removal.

The protective properties of Al_x_Si_1−x_N coatings greatly depend on their microstructure and phase composition, which are substantially affected by the Al/Si ratio. Since AlN and Si_3_N_4_ are considered to be mutually immiscible [37,38], the material deposited at close to equilibrium conditions comprises a mixture of these phases [28]. Nevertheless, in the case of physical vapor deposition, when coatings grow far from equilibrium conditions, Si atoms partially substitute Al atoms in the hexagonal AlN (wurtzite) crystal lattice up to a certain solubility limit (from 4 to 6 at. % [30,39]), which results in the formation of an Al-Si-N solid solution [30]. At higher Si content, the substitutional incorporation of Si in the w-AlN lattice is energetically unfavorable owing to its large distortions, and therefore, the Si_x_N_y_ phase precipitates at AlN grain boundaries. This leads to encapsulation of Al_x_Si_1−x_N crystallites by a Si_x_N_y_ tissue phase [40]. The more there is excess amount of silicon in the coatings, the more silicon nitride precipitates. Therefore, an increase in the Si content is accompanied by a decrease in the grain size to less than 5 nm. That provides an increase in the surface to volume ratio of the grains and, consequently, an increase in the total volume of the tissue phase in which thickness remains nearly constant up to a Si substitution content of 12 at. % [40]. Further increasing of the Si content cannot be accommodated by a decrease in the grain size and, therefore, the Al_x_Si_1−x_N coating acquires amorphous structure.

There is contradictory information about an optimal Al/Si ratio in Al_x_Si_1−x_N coatings that provides their improved performance. Pelisson et al. [30] have reported on the maximum hardness of the coatings with a relatively low Si content (around 10 at. %), while Musil et al. [31] have found that the amorphous Al_x_Si_1−x_N coatings with ~40 at.% of Si exhibit the higher hardness than that of the polycrystalline coatings with a low (<10 at. %) Si content. It should be also noted that whereas the effect of the Al/Si ratio on the mechanical properties of Al-Si-N is adequately studied [30,31,33,34,35], there is only very little data on their thermal stability and oxidation resistance [31,32]. It has been found that both the crystalline Al-Si-N coatings with a low Si content (<10 at.%) and amorphous Al-Si-N coatings with a high Si content (>20 at.%) possessed high oxidation resistance up to temperatures of ~1000 °C and ~1150 °C, respectively [31]. In addition, the amorphous coatings with 23 at. % of Si exhibited high thermal stability up to 1100 °C [32], while the coatings with even higher Si content (~40 at. %) were stable up to ~1150 °C [31]. Thermal stability of crystalline Al-Si-N coatings with a low Si content has not been studied yet. Moreover, in the referred studies Al_x_Si_1−x_N coatings were deposited on Si [30,31,33,34], steel [31,33,34], corundum [31], glass [33] and WC-Co substrates [35], while there is a lack of information about thermal protection and barrier properties of the coatings on Cu substrates. Although theoretical modeling tools such as the Synthetic Growth Concept based on the Density Functional Theory [41] can provide prediction of structure and properties of nanostructured compound coatings, experimental investigations are needed to validate theoretical results. Therefore, the aim of this study is to investigate the effect of the Al/Si ratio on the thermal cycling performance of Al_x_Si_1−x_N coatings deposited on Cu substrates.

## 2. Experimental Part

Al_x_Si_1−x_N coatings were deposited by pulsed bipolar DC magnetron sputtering [42] on stationary copper (99.94% Cu) plates (20 mm × 20 mm × 2 mm) in a mixed Ar + N_2_ reactive atmosphere at a total pressure of 0.3 Pa (the partial pressure of nitrogen was 0.06 Pa) and a substrate temperature of 350 °C. Before the coating deposition the substrates were mechanically polished followed by ultrasonic cleaning in rectified alcohol. In addition, the plates were sputter-cleaned by Ar^+^ ions (at the energy 400 eV and the operating pressure 0.15 Pa) during 20 min to enhance adhesion of the coatings.

Based on the results of earlier studies [30,39,40], an Al-Si mosaic target as well as Al_0.7_Si_0.3_ and Al_0.1_Si_0.9_ alloy targets were used to obtain the coatings with Al-Si-N solid solution, two-phase (Al_x_Si_1−x_N nanocrystallites embedded in the Si_x_N_y_ tissue phase) and amorphous structure, respectively. All targets were 125 mm in diameter. The mosaic target was represented as a pure Al plate with 12 Si circular wafers with a diameter of 15 mm uniformly distributed on the plate along a circle with a radius of a half of that of the plate. The target power density was 9 W/cm^2^; the substrate to target distance was 90 mm. The frequency of current pulses was 50 kHz; the lengths of the negative and positive pulses were 16 and 4 µs, respectively, which corresponded to the 80% duty cycle. The applied substrate bias was –100 V. The deposition rate was approximately 1.5 nm/s and the coating thickness was varied in the range 3–5 µm.

The surface and cross-sectional morphologies as well as the elemental composition of the Al_x_Si_1-x_N coatings were observed with a Quanta 200 3D scanning electron microscope (SEM, FEI, Eindhoven, The Netherlands) equipped with an energy dispersive X-ray spectrometer (EDX). A JEM-2100F transmission electron microscope (TEM, JEOL, Tokyo, Japan) equipped with an EDX detector was employed for microstructural characterization of the samples in the plan-view and cross-section geometries. The structure and phase composition of the coatings were investigated by X-ray diffraction (XRD) in the Bragg–Brentano geometry using an XRD 6000 diffractometer (Shimadzu, Kyoto, Japan) with Co kα radiation. Nanoindentation measurements were carried out with a NanoTest system (Micro Materials Ltd., Wrexham, UK) using a Berkovich indenter. The maximum applied load was 10 mN. The hardness *H* and the elastic modulus *E* of the coatings were determined from load versus displacement curves using the Oliver-Pharr method [43].

Thermal cycling experiments were performed in air using a muffle furnace. The specimens were placed inside the furnace, heated with 30 K/min, annealed during 1 min at a constant temperature of 1000 °C and cooled down at a rate of ~50 K/min to room temperature outside the furnace.

## 3. Results

### 3.1. Characterization of the As-Deposited Coatings

EDX analysis showed that all the coatings are stoichiometric in nitrogen (~50 at. %). The coatings deposited using the mosaic target were represented as Al_0.88_Si_0.12_N. The coatings sputtered from the Al_0.7_Si_0.3_ and Al_0.1_Si_0.9_ alloy targets have a composition of Al_0.74_Si_0.26_N and Al_0.11_Si_0.89_N, respectively. The aluminium content in these coatings is slightly higher than that in the corresponding targets due to its higher sputtering yield than that of silicon. Thus, the Si contents in the coatings are 6, 13 and 44.5 at. % that according to the earlier studies [30,39,40] corresponds to their Al-Si-N solid solution, two-phase (Al_x_Si_1−x_N nanocrystallites encapsulated by the Si_x_N_y_ tissue phase) and amorphous structure, respectively. The latter is in a good agreement with the results of XRD investigations of the coatings which are presented in Figure 1 and their TEM and high-resolution TEM (HRTEM) micrographs shown in Figure 2. It is seen from Figure 1a (curve 1) that, except for the diffraction peaks due to the Cu substrate, only peaks of the AlN hexagonal wurtzite phase (w-AlN) without preferred orientation are visible in the diffraction pattern of the as-deposited Al_0.88_Si_0.12_N coating. It should be noted that the AlN diffraction peaks shift to higher angles. This is because the replacement of Al atoms in the w-AlN lattice by the Si atoms induces its contraction that is shown to be a consequence of charge compensation [44]. The Al_0.88_Si_0.12_N coating is characterized by columnar grains (Figure 2a) with ordered crystalline structure that is clearly visible from the micrograph displayed in Figure 2b. The ordered crystalline structure of the Al_0.88_Si_0.12_N coating is also clearly visible from the micrograph displayed in Figure 2a. The Al_0.74_Si_0.26_N coating exhibits a much-broadened w-AlN peak over the 2θ range from 38° to 44° indicating that the coating tends to become X-ray amorphous due to a very small crystallite size (Figure 1b, curve 1). This exactly matches with the HRTEM image shown in Figure 2c that exhibits small misoriented w-AlN crystallites (3–5 nm in diameter) embedded into the amorphous matrix. Finally, no signals from any crystalline phase except for the Cu substrate are observed in the diffraction pattern of the Al_0.11_Si_0.89_N coating (Figure 1c, curve 1), which means it has a fully amorphous structure. Disordered amorphous structure is also evident from the micrograph of the Al_0.11_Si_0.89_N coating (Figure 2d).

The structural changes of the Al_x_Si_1−x_N coatings caused by increasing the Si content result in substantial variations of their mechanical characteristics (Table 1). The Al_0.88_Si_0.12_N coating is characterized by a rather low hardness and elastic modulus which are similar to the values typical for aluminium nitride coatings [45]. The enhancement of the mechanical characteristics of the Al_0.74_Si_0.26_N coating can be attributed to their nanocomposite microstructure that hinders dislocation motion and thus contributes to the hardening effect. The amorphous Al_0.11_Si_0.89_N coating exhibits lower values of *H* and *E* that can be explained by its disordered structure and the difference in deformation mechanisms. Obviously, the mechanisms typical for polycrystalline coatings such as grain boundary sliding and hindering of dislocation motion by grain boundaries are not relevant in amorphous structures. Thus, the highest hardness value was exhibited by the coating with 13 at. % of Si, which is in good agreement with earlier reported results for magnetron sputtered Al_x_Si_1−x_N coatings [30].

### 3.2. Thermal Cycling of the Coatings

Thermal cycling tests revealed that degradation of the Al_0.88_Si_0.12_N coating became visible after 20 cycles of heating and cooling. Round defects 50–100 µm in diameter, which central part is pressed into the substrate, were formed on the coating surface (Figure 3a,b). The elemental composition of the defects is 93 at. % of copper and 7 at. % of oxygen, i.e., they contain copper that diffuses from the substrate to the surface and partially oxidizes. Outside the defects the average copper content in the coating is less than 6 at. %, which indicates only small changes in its composition. After 25 cycles, local spallation of the Al_0.88_Si_0.12_N coating was observed that occurred either around the defects (Figure 3c) or all over the defect area (Figure 3d). In the former case, the coating spallation resulted from interfacial fracture, while in the latter case, it was due to the propagation of cracks into the substrate that led to tearing substrate fragments and the formation of pits on the specimen surface. XRD measurements revealed disappearance of w-AlN (100) and (101) reflections after 25 thermal cycles, while the intensity of the w-AlN (002) peak increased (Figure 1a, curve 2). In addition, copper oxide peaks are observed in the XRD pattern of the Al_0.88_Si_0.12_N coating.

In contrast, the Al_0.74_Si_0.26_N coating was found to be characterized by poor thermal cycling resistance. This coating was subjected to cracking and spallation already after the first thermal cycle (Figure 4). Therefore, although there was not observed changes in the structure and elemental composition of the coating, its rapid failure makes impossible to estimate its barrier performance against copper diffusion. It should be noted that the reflections of copper oxides appeared in its XRD pattern (Figure 1b, curve 2), however, this can be attributed to oxidation of the substrate surface on the areas of the coating spallation.

Finally, the Al_0.11_Si_0.89_N coating exhibited even poorer thermal cycling performance. Cracking and spallation of this coating was also observed after the thirst thermal cycle (Figure 5a). In addition, extensive copper diffusion into the Al_0.11_Si_0.89_N coating was observed that resulted in changes of its structure and phase composition. As can be seen from a cross-sectional SEM micrograph of the Al_0.11_Si_0.89_N coating given in Figure 5b, columnar structure forms there during annealing. EDX analysis showed that the columnar structure (see point A in Figure 5b) predominantly contains Cu (49 at. %) and Si (48 at. %), i.e., significant copper enrichment of the Al_0.11_Si_0.89_N coating occurred during the first thermal cycle. It should be noted that the copper enrichment is non-uniform across the coating surface. As clearly seen from the EDX maps of the Al_0.11_Si_0.89_N coating surface presented in Figure 6, the copper enrichment is more pronounced in the central part of the fragments bordered by through-thickness cracks, whereas it is not revealed near the fragment edges. The non-uniform distribution of copper-enriched domains also manifests itself in the wave-like variation of the thickness of columnar structure clearly visible in the cross-sectional SEM micrograph of the Al_0.11_Si_0.89_N coating (Figure 5b). In the central part of the fragments, Cu atoms diffuse throughout the Al_0.11_Si_0.89_N coating and accumulate on its surface as crystallites of 1–2 µm in size observed in Figure 5c. EDX analysis showed that the crystallites are comprised of 87 at. % of copper and 13 at. % of oxygen. The observed changes in the elemental composition of the Al_0.11_Si_0.89_N coating is confirmed by XRD measurements (see Figure 1c, curve 2). In addition to copper oxide reflections which were found for the coatings with higher Al/Si ratio, X-ray diffraction reveals the presence of copper silicide Cu_5_Si and silicon dioxide SiO_2_ in the Al_0.11_Si_0.89_N coating. There are also observed diffraction peaks which can be attributed to Al_6_Si_2_O_13_ (mullite) phase that is known to start to form at 950–1000 °C [46].

## 4. Discussion

The results of the thermal cycling tests imply that the main factors that induce degradation of Al_x_Si_1-x_N coatings on Cu substrates during temperature excursions within the range studied are thermal stress arising due to the difference in their coefficients of thermal expansion (CTEs) and diffusion processes in the coating-substrate system. Since, CTE of copper (α_Cu_ = 16.4 × 10^−6^ °C^−1^) is substantially higher than that of Al-Si-N (which CTE α_Al-Si-N_ can vary from $\alpha_{Si_{3}N_{4}}$ = 2.8 × 10^−6^ °C^−1^ to α_AlN_ = 5.3 × 10^−6^ °C^−1^ depending on a Si content), because of rigid bonding at the interface, the elastic strain arises during heating to fit the coating to the substrate:(1)$$\varepsilon =-\left(\alpha_{\mathrm{Al}-\mathrm{Si}-N}-\alpha_{\mathrm{Cu}}\right)\Delta T,$$
where Δ*T* is the change in temperature. During heating, Δ*T* > 0, and the elastic accommodation strain determined by Equation (1) is positive. Therefore, tensile biaxial in-plane stresses are developed in the coating:(2)$$\sigma =\frac{E}{1-\nu }\varepsilon ,$$
where ν is the Poisson’s ratio of the coating. Apparently, these stresses induce cracking of the Al_0.74_Si_0.26_N and Al_0.11_Si_0.89_N coatings already during the first thermal cycle. In the former case, this can be attributed to the highest elastic modulus among the coatings studied, whereas in the latter case it appears to result from combination of the high enough *E* value and the largest thermal expansion due to the highest Si content. In contrast, cracking of the Al_0.88_Si_0.12_N coating did not occur because of its lowest elastic modulus and highest Al content that provides the largest CTE and, therefore, the lowest elastic strain.

It can be thought that the thermal stresses also greatly affect the barrier performance of the coatings against copper diffusion. This is especially true for the Al_0.11_Si_0.89_N coating, where substantial changes of structure and phase composition were revealed already after the first thermal cycle. The latter can be attributed to a high Si content in this coating. Copper is known to react with silicon already at 200 °C [47], whereas diffusion of copper in silicon nitride occurs at temperatures above 550 °C [48,49]. Therefore, during heating up to an annealing temperature of 1000 °C, copper atoms easily diffuse from the substrate into the Al_0.11_Si_0.89_N coating and partly react with silicon to form the metastable copper-enriched Cu_5_Si silicide phase revealed in the XRD-pattern. The phase transformations in the Al_0.11_Si_0.89_N coating are supposed to result in the formation of the columnar structure revealed in its cross-sectional SEM images. Evidently, copper diffusion was more pronounced in the central part of the coating fragments, where the thicker columnar layer is observed. Moreover, some of copper atoms penetrate throughout the coating and react with oxygen to form copper oxide crystallites on the coating surface (Figure 5c). On the other hand, near the edge of the coating fragments the columnar layer is rather thin. The non-uniform thickness of the columnar structure is in good agreement with the expected distribution of in-plane tensile stress developed in the coating during heating. According to the numerical simulation results, after cracking, the tensile in-plane stress component is drastically reduced in a region near the edge of the coating fragments but it remains unrelaxed far from the edge [50]. This implies that that the in-plane tensile stress in the coating could facilitate copper outdiffusion from the substrate.

The Al_0.88_Si_0.12_N coating exhibited the highest diffusion barrier performance among the coatings studied. This can be attributed to the highest fraction of the AlN phase in this coating that not only provides the lowest thermal strain and stress, as discussed above, but also hinders copper diffusion. In particular, Lee et al. reported that AlN effectively blocked the fast pathways for Cu diffusion up to a temperature of 1000 °C [51]. In contrast to the Al_0.11_Si_0.89_N coating, copper diffusion through the Al_0.88_Si_0.12_N coating, that resulted in the formation of the round defects after 20 thermal cycles, apparently occurred only in the areas of defects (pores and microcracks), which arise and coalesce during thermal cycling providing for fast diffusion pathways. This led to mass transfer from the substrate to the coating surface and, as a result, to a local density decrease in the substrate under the defects. The latter favors bending of the coating and its pressing into the substrate on the following cooling. This can be thought of as an indication of high compressive stresses in the Al_0.88_Si_0.12_N coating arising during cooling. The latter is in good agreement with the results of Pélisson, who has reported about a large increase of the compressive stress in annealed Al-Si-N coatings with a Si content above 4 at. % [52]. The pressing of the Al_0.88_Si_0.12_N coating could contribute to the development of compressive stresses in the underlying substrate and, consequently, to enhancement of copper outdiffusion. In addition, the pressing of the coating should lead to arising an internal bending moment and a stress normal to the coating/substrate interface along the perimeter of the bent surface area [46]. The latter is assumed to induce local spallation of the coating around the extrusions on the subsequent thermal cycling.

The results obtained showed that, when deposited on Cu substrates, the crystalline coatings with a low Si content (Al_0.88_Si_0.12_N) exhibited the substantially higher thermal cycling resistance than the two-phase and amorphous coatings with medium (Al_0.74_Si_0.26_N) and high (Al_0.11_Si_0.89_N) Si contents, respectively. These findings are not quite predictable because (i) usually it is believed that amorphous coatings ensure better oxidation resistance than the crystalline ones [53], since the amorphous materials contain no grains and, consequently, no grain boundaries which represent easy paths for oxygen diffusion especially in the case of columnar grains, and (ii) the silicon nitride phase is considered to be more stable than the metal nitride phase [53]. However, earlier experiments [31] have showed that the crystalline Al-Si-N coatings containing a large amount of Al also exhibit good oxidation resistance that is due to the presence of free Al atoms in the coating from which easy oxidation results in the formation of a dense protective Al_2_O_3_ surface layer preventing the fast penetration of oxygen into the bulk of the coating. Moreover, the present study revealed that the degradation of Al-Si-N coatings deposited on Cu substrates primarily occurred due to cracking and spallation of the coatings as well as their phase transformations caused by outdiffusion of Cu atoms from the substrate rather than to coating oxidation. In such circumstances the key factors governing the thermal cycling performance of the coatings are their thermomechanical and barrier properties, which provide lower thermal strains during temperature excursions and prevent the outdiffusion of substrate elements. It is evident that the Al_0.88_Si_0.12_N coating best meets the requirements.

## 5. Conclusions

The Al/Si ratio in Al_x_Si_1−x_N coatings magnetron sputtered on Cu substrates was found to greatly affect their structure, mechanical properties and thermal cycling resistance. The amorphous coating with a high Si content (Al_0.11_Si_0.89_N) was subjected to cracking and spallation just after the first cycle of annealing at a temperature of 1000 °C. Rapid degradation of the coating was due to tensile thermal stresses arising on heating owing to difference in CTEs of the coating and the substrate as well as to changes in elemental and phase compositions of the coating caused by copper diffusion from the substrate. The coating with an intermediate Si content (Al_0.74_Si_0.26_N) comprised of AlN nanocrystallites embedded in the amorphous Si_x_N_y_ matrix exhibited the better barrier performance against copper diffusion from the substrate. However, it could not be correctly estimated because the high elastic modulus of this coating resulted in high thermal stresses that caused its cracking spallation after the first thermal cycle. Finally, the coating with a low Si content (Al_0.88_Si_0.12_N), the main phase of which was polycrystalline AlN, was characterized by the highest thermal cycling resistance, which can be attributed to its rather low elastic modulus that ensured a substantial decrease in the thermal strain and stress. This coating also exhibited the highest diffusion barrier performance, which was due to the large fraction of AlN phase which effectively blocked the fast pathways for Cu diffusion. As a result, degradation of the Al_0.88_Si_0.12_N coating became visible only after 20 thermal cycles as round defects on the specimen surface, which formed due to copper diffusion from the substrate through voids formed due to thermomechanical load. Local spallation of the Al_0.88_Si_0.12_N coating in the defect regions starts after 25 cycles of annealing. The results obtained testify that the Al_x_Si_1−x_N coatings with low Si content can provide rather high thermal cycling performance when deposited on Cu substrates. Considering that the main phase in these coatings is AlN, the thermal conductivity of which can reach 80% of that of copper, they have promise for the protection of copper components used for heat removal.

## Figures and Tables

**Figure 1 materials-12-02249-f001:**
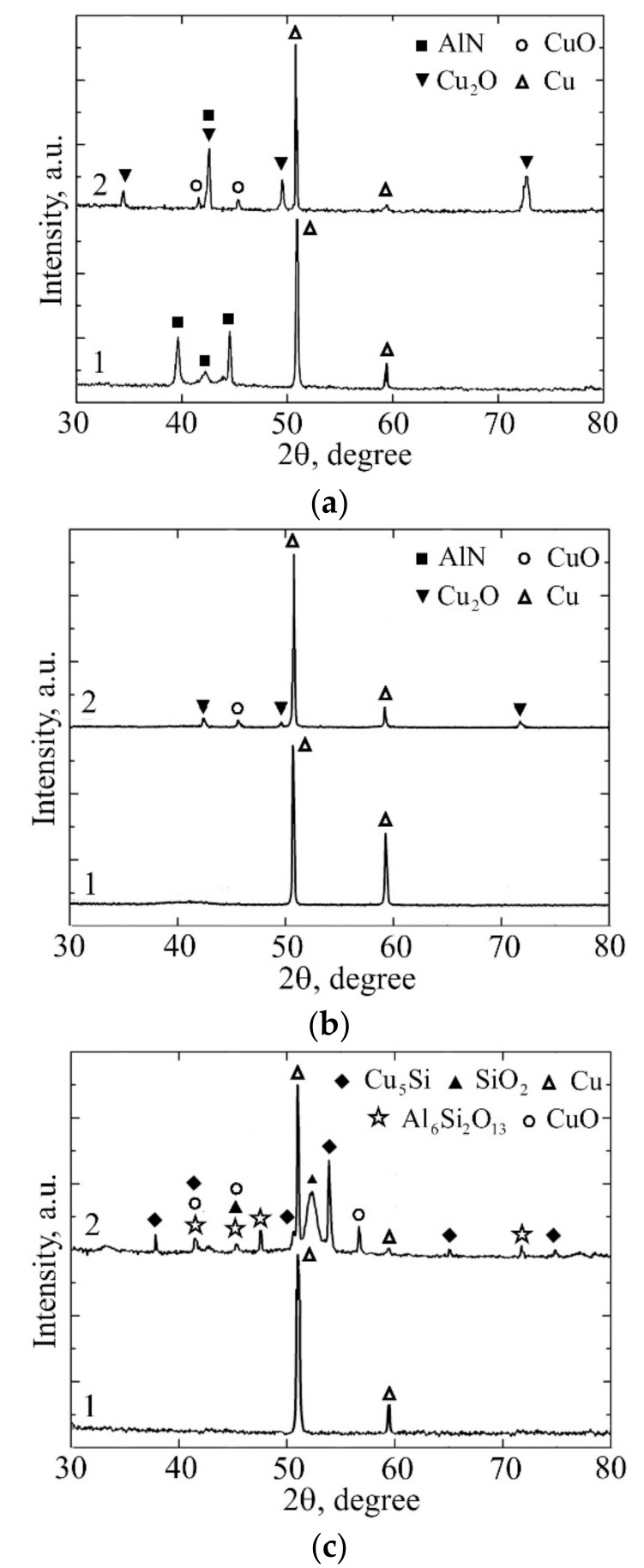
XRD patterns of Al_0.88_Si_0.12_N (**a**), Al_0.74_Si_0.26_N (**b**) and Al_0.11_Si_0.89_N (**c**) coatings. The curves labeled as (1) correspond to the as-deposited coatings, while those designated as (2) belong to the coatings subjected to 25 (**a**) and 1 (**b**,**c**) thermal cycles.

**Figure 2 materials-12-02249-f002:**
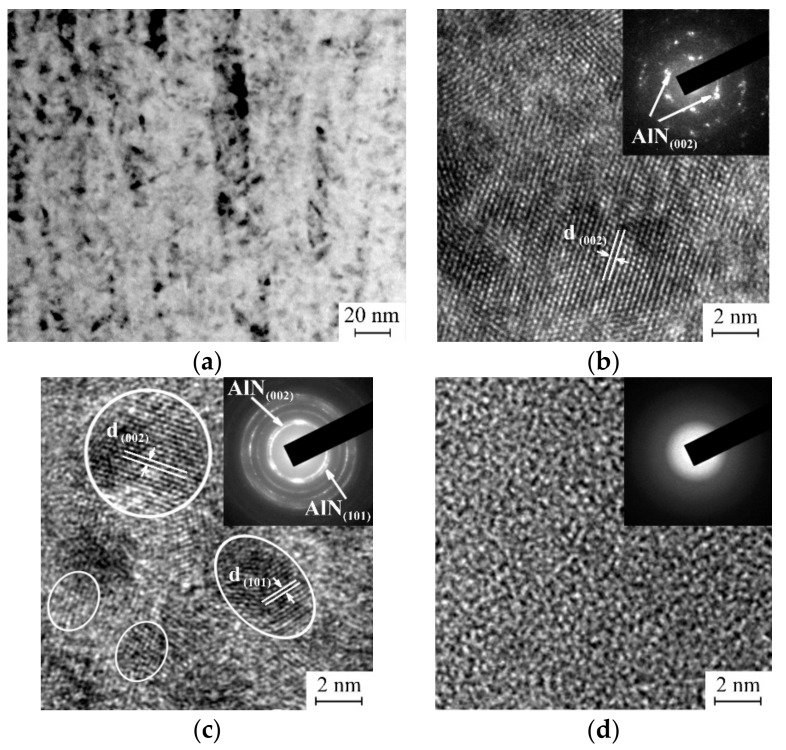
HRTEM micrographs and corresponding selected area electron diffraction patterns (top right insets) of as-deposited Al_0.88_Si_0.12_N (**a**,**b**), Al_0.74_Si_0.26_N (**c**) and Al_0.11_Si_0.89_N (**d**) coatings. For clarity the nanocrystallites in (**c**) are outlined by white dashed contours.

**Figure 3 materials-12-02249-f003:**
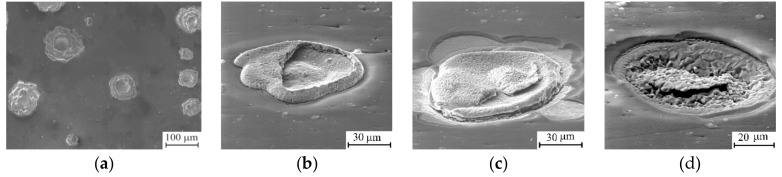
SEM micrographs of Al_0.88_Si_0.12_N coating after 20 (**a**,**b**) and 25 (**c**,**d**) thermal cycles.

**Figure 4 materials-12-02249-f004:**
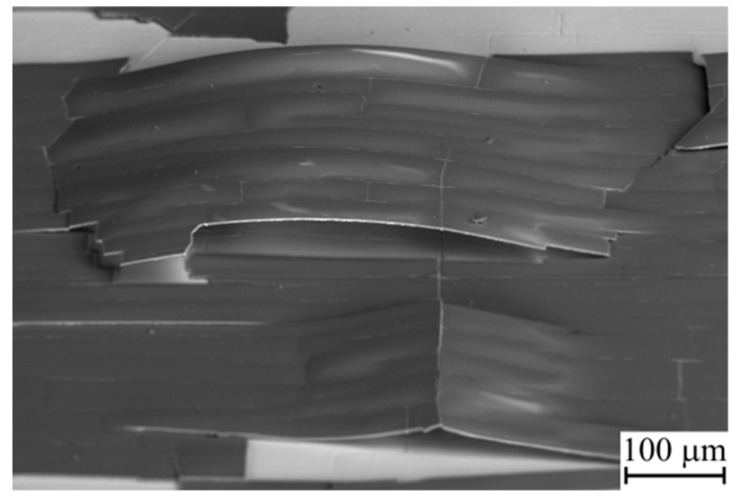
SEM micrographs of Al_0.74_Si_0.26_N coating after 1 thermal cycle.

**Figure 5 materials-12-02249-f005:**
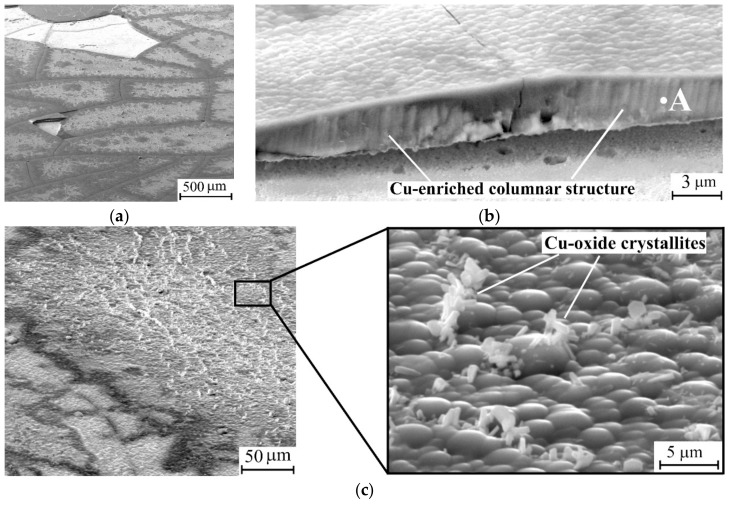
SEM micrographs showing a surface (**a**,**c**) and a cross-section (**b**) of Al_0.11_Si_0.89_N coating after 1 thermal cycle.

**Figure 6 materials-12-02249-f006:**
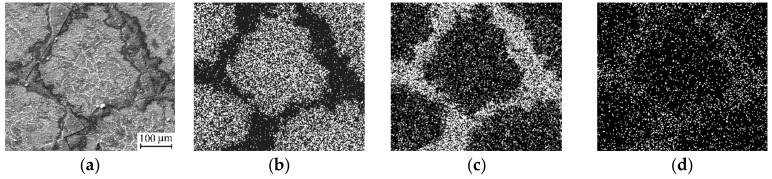
A SEM micrograph (**a**) and corresponding EDX maps of Cu (**b**), Si (**c**) and Al (**d**) for an Al_0.11_Si_0.89_N coating after 1 thermal cycle. White and black colors correspond to high and low elemental contents, respectively.

**Table 1 materials-12-02249-t001:** Hardness and elastic modulus of Al-Si-N coatings.

Coating	*H*, GPa	*E*, GPa
Al_0.88_Si_0.12_N	10.9 ± 1.8	135 ± 12
Al_0.74_Si_0.26_N	24.6 ± 2.1	228 ± 11
Al_0.11_Si_0.89_N	20.2 ± 2.2	180 ± 9

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
