# Peer review of "Improvement of Thermal Cycling Resistance of AlxSi1−xN Coatings on Cu Substrates by Optimizing Al/Si Ratio"

_materials, 2019, doi:10.3390/ma12142249_

Round 1
Reviewer 1 Report
The article reports on a systematic investigation of thermal cycling resistance of AlSiN coatings on Cu substrate. This is a very interesting paper that ultimately should be published. At present, however, the manuscript requires substantial re-working. Prior to publication, the paper should be completed and modified according to the following comments.
Comments and recommendation
Overall major comments:
1. At the beginning, the authors introduce the reader to the AlSiN coatings with an emphasis on structural features and phase composition. However, details on its phase stability at elevated temperatures would be more relevant to the general aim of the manuscript. Moreover, the authors should compare the thermal cycling resistance of the coatings in terms of their microstructure. It was observed earlier that typically amorphous coatings have better oxidation resistance than coatings with crystalline structure (see for instance 10.1016/j.surfcoat.2012.05.073) but this is not the case in the present study. Please, discuss.
2. Discussion. Lines 221-253. The presented experimental results do not confirm directly that the main factors that induce degradation of the coatings are thermal stress arising due to the difference in their coefficients of thermal expansion and diffusion processes in the coating‐substrate system. There are no measurements of stresses and diffusion of elements during thermal cycling. Therefore, the text in the section ”Discussion” is nothing more than an interpretation of the results. Therefore, you should avoid such expressions like “The investigations performed confirmed”, “The results obtained showed that” etc. The section “Discussion” should be re-written keeping this in mind.
Specific comments on the results:
Lines 50-51.It is worth mentioning here the novel nitrides based on high entropy alloys, for instance, 10.1016/j.matlet.2018.07.048.
There is no information on Nitrogen concentration in the coatings. Please, add this.
Details on the microstructure of the Al0.88Si0.12N coating should be provided. It is not clear whether the coating is polycrystalline with columnar microstructure or fine-grained microstructure.
Fig.1. The figures, symbols and text should be increased.
Table 1. It is worth to add some values earlier reported for magnetron sputtered AlSiN coatings with the same phase composition.
Page 6. Lines 196-198. These EDS results should be provided in the manuscript. The caption on Fig.5b “Cu-enriched columnar structure” remains unjustified without them.
Fig.5c. What does the caption “d” means? It should be removed or explained.
Lines 247-250, 267-273. These statements have not been justified by experimental evidence. Please, re-formulate to be clear.
The article contains very valuable experimental outputs, but they should be analyzed more detailed and interpreted correctly. Therefore, the major revision is necessary.
Author Response
Dear reviewer,
we would like to thank you for taking your time to consider our manuscript “The effect of Al/Si ratio on thermal cycling resistance of AlxSi1-xN coatings on Cu substrates” and for valuable comments and suggestions that helped us to improve the manuscript.
Our responses to the comments are listed below following each specific comment. We also addressed the comments and suggestions in the revised manuscript. All the changes made in the revised manuscript were colored in yellow.
1. At the beginning, the authors introduce the reader to the AlSiN coatings with an emphasis on structural features and phase composition. However, details on its phase stability at elevated temperatures would be more relevant to the general aim of the manuscript. Moreover, the authors should compare the thermal cycling resistance of the coatings in terms of their microstructure. It was observed earlier that typically amorphous coatings have better oxidation resistance than coatings with crystalline structure (see for instance 10.1016/j.surfcoat.2012.05.073) but this is not the case in the present study. Please, discuss.
Response: The details of the earlier works concerned with thermal stability and oxidation resistance of Al-Si-N coatings are added into the section “Introduction” (Lines 84-92). Discussion of the thermal cycling resistance of the coatings in terms of their microstructure is provided in the section “Discussion” of the revised manuscript (Lines 290-307).
2. Discussion. Lines 221-253. The presented experimental results do not confirm directly that the main factors that induce degradation of the coatings are thermal stress arising due to the difference in their coefficients of thermal expansion and diffusion processes in the coating‐substrate system. There are no measurements of stresses and diffusion of elements during thermal cycling. Therefore, the text in the section ”Discussion” is nothing more than an interpretation of the results. Therefore, you should avoid such expressions like “The investigations performed confirmed”, “The results obtained showed that” etc. The section “Discussion” should be re-written keeping this in mind.
Response: The section “Discussion” was re-written according to the reviewer’s recommendations.
3. Lines 50-51.It is worth mentioning here the novel nitrides based on high entropy alloys, for instance, 10.1016/j.matlet.2018.07.048.
Response: The references on nitride coatings based on high entropy alloys are mentioned in the revised manuscript (Lines 51-52).
4. There is no information on Nitrogen concentration in the coatings. Please, add this.
Response: Information on Nitrogen concentration in the coatings is presented in the revised manuscript (Line 135).
5. Details on the microstructure of the Al0.88Si0.12N coating should be provided. It is not clear whether the coating is polycrystalline with columnar microstructure or fine-grained microstructure.
Response: Details on the microstructure of the Al0.88Si0.12N coating are presented in the revised manuscript (Fig. 2a and Lines 149-150).
6. Fig.1. The figures, symbols and text should be increased.
Response: The XRD-patterns, symbols and text presented in Fig. 1 were increased.
7. Table 1. It is worth to add some values earlier reported for magnetron sputtered AlSiN coatings with the same phase composition.
Response: Along with the phase and elemental compositions the mechanical characteristics of magnetron sputtered coatings are greatly affected by the deposition parameters. As shown in the earlier works the coatings with the same elemental composition can be characterized by very different values of hardness, elastic modulus, etc. For example, it has been found that depending on the deposition conditions the hardness of Al-Si-N coatings with a low (≤10 at.%) Si content varied from 10.3 to 26.7 GPa, and their reduced elastic modulus changed from 149 to 209 GPa [10.1016/j.surfcoat.2016.05.054]. Therefore, it seems to be confusing for the reader to compare our results with the earlier reported values obtained for the Al-Si-N coatings sputtered in different conditions.
8. Page 6. Lines 196-198. These EDS results should be provided in the manuscript. The caption on Fig.5b “Cu-enriched columnar structure” remains unjustified without them.
Response: To justify the caption on Fig.5b “Cu-enriched columnar structure” we indicate on Fig. 5b point A, where the EDS date were obtained.
9. Fig.5c. What does the caption “d” means? It should be removed or explained.
Response: Figs. 5c,d were modified and the caption “d” was removed.
10. Lines 247-250, 267-273. These statements have not been justified by experimental evidence. Please, re-formulate to be clear.
Response: These fragments were re-written in the revised manuscript to be clear (Lines 265-270 and 284-289).
Reviewer 2 Report
This paper is dedicated to the effect of the elemental composition of AlxSi(1-x)N coatings deposited on Cu substrates by magnetron sputtering on structure, mechanical properties and thermal cycling performance. The study employs the appropriate methodology which is accordingly discussed and explained.
This research is timely (also due to the increased interest in Al‐Si‐N coatings/solid solutions) and quite valuable, and the authors addressed the topic in a relatively original and at the same time easy to perceive way. Magnetron sputtering is a technique with excellent prospects for this type of compounds.
There are well-presented figures, as well as concise discussion which makes the manuscript an informative read.
There are some aspects of this good manuscript that need revision; thus, it is acceptable for publication after minor revision:
1: Introduction is written well. There are theoretical modeling tools such as Synthetic Growth Concept (DFT-based) that are applicable to compounds grown by similar technique, i.e., magnetron sputtering, mention access to such tools will add to the substance of this manuscript, e.g.,
Chemical Physics Letters Volume 506, (2011) Pages 86-91
2: The title is too descriptive, the authors may thing of a more concise and attractive title.
3: The English of the paper is good, but still needs some spell-checking and stylistic improvements “even poorer“ instead of “even more poor” – page 6, and so on.
4: The interplay between structure, elastic properties and temperature is discussed in the manuscript in the spirit of the concrete experimental results, without trying to refer to more general principles or dependencies. May be for the seek of explaining the results from more general point of view, the authors could try to give a different, more general interpretation of the dependencies they have found.
Author Response
Dear reviewer,
we would like to thank you for taking your time to consider our manuscript “The effect of Al/Si ratio on thermal cycling resistance of AlxSi1-xN coatings on Cu substrates” and for valuable comments and suggestions that helped us to improve the manuscript.
Our responses to the comments are listed below following each specific comment. We also addressed the comments and suggestions in the revised manuscript. All the changes made in the revised manuscript were colored in yellow.
1. Introduction is written well. There are theoretical modeling tools such as Synthetic Growth Concept (DFT-based) that are applicable to compounds grown by similar technique, i.e., magnetron sputtering, mention access to such tools will add to the substance of this manuscript, e.g., Chemical Physics Letters Volume 506, (2011) Pages 86-91
Response: The proposed reference to the Synthetic Growth Concept was mentioned in the revised manuscript (Lines 95-98).
2. The title is too descriptive, the authors may thing of a more concise and attractive title.
Response: The title was modified according to the reviewer’s recommendation.
3. The English of the paper is good, but still needs some spell-checking and stylistic improvements “even poorer“ instead of “even more poor” – page 6, and so on.
Response: The manuscript was subjected to careful proofreading. All the changes are highlighted in yellow through the text of the revised manuscript.
4. The interplay between structure, elastic properties and temperature is discussed in the manuscript in the spirit of the concrete experimental results, without trying to refer to more general principles or dependencies. May be for the seek of explaining the results from more general point of view, the authors could try to give a different, more general interpretation of the dependencies they have found.
Response: The section “Discussion” was re-written according to the reviewer’s recommendations.
Round 2
Reviewer 1 Report
The authors have improved their manuscript following my comments and recommendations. Now, I can recommend the manuscript for publication in the journal.